# Effects of Cryogenic Storage on Human Amnion Epithelial Cells

**DOI:** 10.3390/cells9071696

**Published:** 2020-07-15

**Authors:** Raghuraman C. Srinivasan, Stephen C. Strom, Roberto Gramignoli

**Affiliations:** Department of Laboratory Medicine, Division of Pathology, Karolinska Institutet, 141 52 Stockholm, Sweden; ragsyrazor@gmail.com (R.C.S.); stephen.strom@ki.se (S.C.S.)

**Keywords:** placenta, amnion membrane, cell-based therapy, cryopreservation, cell transplant

## Abstract

Perinatal stem cells and epithelial cells isolated from full term amnion membrane, in particular, have attracted interest over the last decade, as a promising source of multipotent cells for cellular therapies. Human amnion epithelial cells (hAEC) have been used to treat monogenetic liver disease such as maple syrup urine disease or fibrosis of the liver in preclinical studies. In most studies xeno-transplants of hAEC were conducted without providing immunosuppression to recipients, reflecting the tolerogenic properties of hAEC. For many cell types, successful cryopreservation is critical for providing a readily available, off-the-shelf product. In this study, hAEC were isolated from full-term human placenta from 14 different donors, cryopreserved using a protocol and reagents commonly adopted for epithelial cell preservation. The cells were analyzed in terms of survival, recovery, and homogeneity, profiled for surface markers characteristic of epithelial, mesenchymal, endothelial, or hematopoietic cells. There were no significant differences observed in the percentage of cells with epithelial cell markers before and after cryopreservation. The relative proportion of stromal and hematopoietic cells was significantly reduced in hAEC preparations after cryopreservation. The expression of stem cell and immunomodulatory molecules were confirmed in the final product. Since multipotent cells are readily available from full-term placenta, this novel cell source might significantly increase the number of patients eligible to receive cellular therapies for liver and other diseases.

## 1. Introduction

Cellular therapeutics have potential utility for a large number of unmet medical needs. Large-scale clinical-grade isolation and subsequent banking of stem cells has been the backbone of the field of regenerative medicine. Successful translation of cell-based therapies frequently requires successful long-term storage under conditions that maintain the safety and efficacy of the final cell product. Cell cryopreservation is an established procedure for several clinical therapies, as decades of bone marrow and cord blood hematopoietic stem cell transplants have shown. Nevertheless, epithelial cell-based treatments, including hepatocytes, have suffered from limited reanimation after cryogenic procedure, with considerable loss of viability and function. 

Human amnion epithelial cells (hAEC) are unique stem cells isolated from perinatal tissues. They form a thin epithelial layer lined up in the amniotic cavity surrounding the fetus during gestation. It is perhaps hAEC origin that makes these cells unique and attractive for interventional treatments [1]. The generation of the amnion epithelial lineage occurs prior to the specification of the three germ layers. This unique place in development, between the pluripotent cells of the epiblast and the specification of the three germ layers, may explain the lineage plasticity to hAEC, a property not shared with other perinatal cells.

It has been reported that hAEC differentiate into different lineages and cell types, including ectoderm (neural cells) [2], endoderm (hepatocyte-like cells [3,4,5], and insulin producing islet-like cells) [4,6], as well as mesoderm lineage [7]. In preclinical studies, hAEC have been used to treat stroke, multiple sclerosis [8], cardiac degenerative diseases [7], type 1 diabetes [9], pulmonary [10,11,12], and metabolic liver diseases [13,14,15]. Amnion products have been shown effective in reversing fibrosis in models of cirrhosis in mice [16] and rats [17]. Recently, a 2-year report described safety and efficacy of hAEC clinical infusions in preterm babies with bronchopulmonary dysplasia [18]. 

In all xeno-transplants and clinical infusion, allogenic human cells have been used with no sign of rejection. In addition to their plasticity and tissue remodeling, hAEC have been reported to possess immune-modulatory properties that makes them appealing for cellular therapy. Human amnion epithelial cells are poorly recognized by the immune system, following allogenic [18,19], and even xeno-transplants [9,10,11,12,13,14,15,16]. This immune privilege is partly due to their uniquely regulated expression of human leukocyte antigens (HLA): with the absence of HLA class II antigens or costimulatory molecules [20], low levels of the polymorphic HLA class 1 [20], and constitutive expression of non-polymorphic HLA-G [21,22]. Amnion-derived cells have been shown to have powerful immunomodulatory properties, crucial for evading an immune mediated rejection of the developing embryo by the maternal system, and in support of allogenic transplantation of hAEC in immune competent animals without immunosuppression [9,10,11,12,13,14,15,16,21]. Preclinical studies report a re-education of the host immune-system into a more tolerogenic and regenerative setting [23,24,25]. Finally, hAEC are not immortal [4], and not tumorigenic when transplanted [4,14,26]. 

The current study is part of a multi-step investigation of safety and efficacy in consideration of moving the hAEC therapy into clinical practice. Recently, standardized reagents and reproducible procedures, in accordance with current good manufacturing procedure (GMP) requirements, have been standardized [27]. Criteria for characterization and release of the final product [27], in addition to biodistribution analysis, have validated an efficient and safe route of administration for qualified hAEC batches into the hepatic parenchyma [28]. 

In anticipation that the transplant product used in any future clinical trial will be previously cryopreserved cells, the current study was undertaken to determine if there are any significant differences between the populations initially isolated from the amnion membranes, and the cells recovered following the cryopreservation procedure. Cell surface markers characteristic of different cell types were investigated both before and after cryopreservation. Additional pathways thought to be critical to hAEC engraftment and long-term function following transplantation were investigated in cryopreserved lots of hAEC to determine if differences could be identified that might provide insight into cases that might be more useful for transplantation as opposed to others.

## 2. Materials and Methods

### 2.1. hAEC Isolation Procedure

The human placenta was procured from uncomplicated full-term cesarean resection from healthy mothers. The placenta was received from Karolinska Institute Hospital, Stockholm under ethical permit number: 2015/419-34/4. Signed, informed consent was provided by the mother to use the sample for research purposes. All the pathogen-positive deliveries (including HBV, HCV, syphilis, and HIV) were excluded, and current revised exclusion criteria includes positiveness for COVID-19. No information regarding human donors (mother and newborn child) was transmitted or diffused, and the details were de-personalized, in order to respect privacy. Placentae were delivered and processed within 3 h post-partum. A protocol for hAEC isolation was previously described [27]. Briefly, the amnion membrane was removed from the inner surface of the placenta and washed multiple times with Ringers solution (Baxter, Sweden) and plasmalyte solution (Baxter, Norfolk, UK) to remove blood. The amnion membrane was digested using TrypLE 10× (Gibco, Grand land, NY, USA) for 30 min at 37 °C to primarily release epithelial cells from the amnion membrane. The dispersed hAEC were collected by centrifugation and filtered through a 100 µm cell strainer. Cell viability was determined by the Trypan Blue exclusion (TBE) method (ThermoFisher, Waltham, MA, USA). 

### 2.2. Cryopreservation Procedure

hAEC were not maintained in culture or selected ex vivo prior to freezing. Following cell isolation, hAEC were immediately re-suspended in University of Wisconsin solution (UW; BEL GEN 1000, Lissieu, France) supplemented with 10% dimethyl sulfoxide (DMSO; Sigma-Aldrich, MO, USA), at a cell density of 10 million viable cells/mL. Every cell batch was aliquoted into 1.5 mL cryotubes (Corning, NY, USA) and transferred in a controlled freezing container that lowered the temperature by 1 °C per minute when stationed in a −80 °C freezer (Biocision, Larkspur, CA, USA). The cells were stored in the vapor phase of a liquid nitrogen storage tank for a minimum of 30 days and a maximum of 36 months before analysis. 

### 2.3. Thawing Procedure

The cryovials were removed from the liquid nitrogen tank and rapidly thawed by partial emersion in a water bath maintained at 37 °C, until small ice crystals remain (commonly for 80–120 s). Cells were diluted into 10 volumes of ice cold plasmalyte solution (Baxter), supplemented with 10% bovine calf serum, and centrifuged at 250 × *g* for 5 min. Cell pellet was resuspended in cold plasmalyte and filtered through a 100 µm cell strainer. Cell viability and recovery were determined by TBE method. 

### 2.4. Flow Cytometry Analysis

The heterogeneity of the cell suspension was evaluated based on surface markers quantified by fluorescence-activated cell sorting (FACS). Both freshly isolated and cryopreserved hAEC were incubated with monoclonal antibodies directed against cell-specific surface protein, properly diluted in PBS solution and incubated for 30 min at 4 °C. The human-specific antibodies included in the study were CD326 (epithelial cell adhesion molecule, EpCAM; clone-HEA-125; Miltenyi Biotech); CD31 (PECAM1; clone WM59), CD44 (HCAM; clone G44-26), CD45 (clone-T29/33), CD49f (alpha 6 integrin subunit; clone-GoH3), CD105 (endoglin; clone-SN6; all from BD Biosciences, San Jose, CA, USA). All six monoclonal antibodies were directly conjugated with one of three specific dyes, fluorescein isothiocyanate (FITC) or phycoerythrin (PE) or allophycocyanin (APC) to perform multilineage evaluation on the same suspension. Corresponding isotype controls were also analyzed. The cells were washed and fixed with 2% BD™ stabilizing fixative (BD Biosciences) for 10 min at room temperature. Cells were washed and re-suspended in ice cold PBS, and analyzed on a FACSCanto (BD Biosciences) using FlowJo™_v10 software. 

### 2.5. Gene Profiling by qPCR

Thawed hAEC were lysed in Trizol™ solution (Life Tech, Carlsbad, CA, USA) and total RNA was isolated according to the manufacturer’s instructions. Total RNA was converted to complementary DNA using high capacity cDNA kit (Life tech, Carlsbad, CA, USA). Gene expression was assessed using TaqMan assays for DLK-1 (HS00171584), MMP2 (HS1548728), MMP3 (HS00968305), MMP7 (HS1042812), MMP8 (HS01029057), MMP9 (HS00957562), MMP 12 (HS00159181), MMP13 (HS00942591), TIMP1 (HS01092512), TERT (HS00972650), OCT4 (HS04260367), NANOG (HS04260366), SOX2 (HS01053049), CD73 (HS00159686), CD39 (HS00969559), CD38 (HS01120071), IDO (HS00984148), HLA-G (HS00365950), HLA-E (HS03045171), HLA-F (HS04185703). Reactions were run in duplicate with human cyclophilin A (PPIA) (Hs99999904_m1) as a house keeping gene as control for all experiments. Calculation of relative levels of expression were done according to the comparative Ct-method as follows: 2^(−ΔCt)^, where ΔCt = (*Ct* gene of interest − *Ct* internal control Cyclophilin). *Ct* values for the gene of interest 35 or higher were considered as unreliable and ignored from the calculation.

### 2.6. Statistical Analysis

Statistical differences were determined by paired *t*-test; *p* < 0.05 was chosen as the minimum level of significance. Results are presented as histograms showing data plots, mean ± standard deviation. All data were analyzed by GraphPad Prism software (version 6.0, GraphPad Software Inc., San Diego, CA, USA).

## 3. Results

Fourteen human placentae were collected and generated cell suspensions characterized by limited variability (16 ± 7 million viable cells/gr of processed tissue). Cell viability measured immediately after isolation was 90% ± 4% (n = 14). When cells from the same 14 cases were thawed months to years later, the average cell viability was significantly lower (78% ± 5%; *p* < 0.0001; Figure 1). Ten million viable hAEC were initially cryopreserved. On average, a lower number of cells were recovered (6.5 ± 1.1 million/mL), corresponding to 55–95% of the initially cryopreserved cells. Cases characterized with the highest viability post-cryogenic procedure did not always result in highest cell recovery (Figure 1). 

### 3.1. Quality Control of Cryopreserved Cell Products

There is some heterogeneity in the types of cells recovered from amnion membrane depending on the isolation procedure and the reagents used. Preparations of hAEC are generally characterized for the presence of static surface markers to identity different cell types in amnion-derived cell product [27]. FACS analysis was performed on each hAEC case, before and after cryopreservation (Figure 2). The presence of specific epithelial cell markers (CD49f and CD326) was confirmed on the majority of isolated cells (Figure 2B). The average expression of CD49f on hAEC was unaffected by cryopreservation procedure and it was present on 99% ± 1% of freshly isolated and cryopreserved cells (*p* = 0.1386) (Figure 2B). Similarly, the expression of another epithelial marker CD326 was non-significantly different between fresh (88% ± 8%) and cryopreserved cells (91% ± 6%; *p* = 0.0939) (Figure 2B).

Digestion of amnion membrane to deliver epithelial cells could result in stromal cell release from the inner layer of the amnion membrane. The presence of amnion-derived mesenchymal stromal cells (MSC) was investigated with common stromal markers, CD105 and CD44 [29]. Approximately 50% of the hAEC preparations contained a small amount of cells with MSC surface markers. Cells positive for CD44 were found on 3/14 preparations, on an average of 0.5% ± 0.8% of the total cell number (Figure 2C), while CD105 was expressed in 9/14 preparations on 1.25 ± 1.6% of the cells. After cryopreservation, stromal marker positive cells were below the limits of detection on 9/14 hAEC (Figure 2C): with 0.1% ± 0.4% (*p* = 0.173) and 0.6% ± 1.4% (*p* = 0.0128) of the total cells remaining positive for CD44 and CD105, respectively.

Hematopoietic or endothelial cells may be present in the final cell suspension from residual contamination with blood or placental vascular tissue. The presence of CD45, a receptor linked to protein tyrosine phosphatase present in cells of the hematopoietic lineage, was detected in 9/12 of hAEC preparations immediately after isolation at an average level of 1.2% ± 1.4% cells, but greatly reduced after cryopreservation and washing steps, where only four preparations still contained an extremely low number of CD45 positive cells (0.3% ± 0.4%; *p* = 0.0146) (Figure 2D). The endothelial cell marker (CD31) was undetectable in all 14 cases, after isolation or cryopreservation (data not shown). 

### 3.2. Immunomodulatory Molecules

Mismatches in human leukocyte antigens (HLA) expression are recognized by immune cells and generally induce rejection. The expression of classical HLA-Ia antigens (HLA-A, -B, -C) and the HLA-II were quantified (Figure 3A). Expression of HLA-Ia was detected on an average of 39% ± 18% of the cells, with a variable range of expression in different cryopreserved hAEC cases (10–55%). As previously reported [20], human leukocyte antigens class II was negative on all hAEC, before and after cryopreservation. 

Amnion epithelial cells have been reported to express characteristic immunomodulatory molecules, such as HLA class Ib [21,22]. These and other surface proteins have been ascribed as mediators in immune-recognition and reactions against innate or adaptive immune cells. The presence of HLA-G on hAEC after isolation was confirmed using a specific antibody, and HLA-G expression was maintained after cryopreservation on 79% ± 9% of the cells (range, 66–91%) (Figure 3A). HLA-Ib expression by hAEC was investigated by transcriptome analysis: HLA-G, -E, and -F forms were detected in all the hAEC cases analyzed, with HLA-E expression approximately 100-fold higher than HLA-G and HLA-F (Figure 3B). Recent studies indicate that ecto-enzymes, such as adenosinergic enzymes CD39 and CD73 play an activate role in modulation of the immune system [25]. Both HLA molecules and ecto-enzymes CD39 and CD73 (30% ± 7% and 88% ± 7%, respectively) were expressed on all 14 cases of hAEC after thawing (Figure 3A). As with HLA-Ib, cryogenic procedure does not significantly affect expression of ecto-nucleotidase enzymes on hAEC (Figure 3B).

Finally, indoleamine-pyrrole 2,3-dioxygenase (IDO), another enzyme involved in immune modulation and immune tolerance by limiting T-cell function and described in immunomodulatory cells, was analyzed but undetectable in all hAEC cases (data not shown).

### 3.3. Pluripotency Genes

Multipotency is another important characteristic ascribed to hAEC [2,17]. The expression of the transcription factors OCT-4, NANOG, and SOX2 are critical in the maintenance of pluripotency, and all three transcription factors were expressed in all hAEC cases (Figure 4A). Another transmembrane protein evolutionarily conserved and responsible for several developmental processes such as cellular fate determination and terminal differentiation, delta-like 1 homolog (DLK-1), was found to be expressed in all preparations (Figure 4A). Finally, telomerase reverse transcriptase (TERT), a component of telomerase complex responsible for long-term survival and repetitive cycles of cell replication was undetectable in all hAEC cases (data not shown).

### 3.4. Cell Engraftment Enzymes

Transplanted epithelial cells require integration into parenchyma to establish cell-to-cell and cell-to-ECM interaction supporting their survival and growth. Integration and nidation is in part facilitated by the secretion of matrix metalloproteinases (MMP), enzymes that degrade ECM and enhance cell adhesion, migration, and proliferation. Several MMPs were analyzed in the hAEC cases. MMP2, MMP3, and MMP9 were expressed in all cases (Figure 4B). Additional isoforms, MMP7, MMP8, and MMP13, were also analyzed but were undetectable in all hAEC cases (Figure 4B). Low level expression of MMP12 was detected in one case, however, it was undetectable in the other 13 cases. Finally, the tissue inhibitor of metalloproteinases TIMP1 was highly expressed in all preparations (Figure 4B). 

## 4. Discussion

Several liver diseases have been treated with liver cell transplants [30,31,32], although human hepatocyte transplants are limited in part by the paucity of organs for cell isolation [31]. Recent studies support the use of hAEC transplants as an alternative for hepatocytes [5,13,14,15,23]. Transplantation of hAEC has been shown to correct amino acids and neural transmitter abnormalities in mouse models of PKU and MSUD [13,14,15], and to reverse the lethal effects of acute liver failure [15,16,33]. The studies with mice, like the human patient transplants, were conducted on subjects that have normal, active immune systems and hAEC were transplanted without providing immunosuppression. Yet, hAEC engrafted without evidence of immune recognition and rejection [13,16,33]. 

As cell therapies become more feasible, the demand for a constant supply of quality cells becomes increasingly important. An optimized enzymatic isolation method has been proved to generate a large amount of hAEC from full-term placentae, and the efficiency in cryopreservation with simple, controlled conditions may fulfill the growing demand for hAEC transplantation. In this study, hAEC were harvested from 14 different donors based on a protocol using entirely clinical grade reagents [27]. The formulation of reagent to thaw hAEC has originally included additive of animal origin (FBS). Such supplement has been replaced with human albumin in revised procedure, and matched reliability and efficiency (as validated in several hAEC batches, including cases here analyzed, data not shown). The quality of cell product is regularly characterized based on the presence of static cell markers (as regularly performed with other cellular therapies). The lack of need for a pre-step based on in vitro selection allowed to immediately direct hAEC to cryogenic preservation, where they were banked for several weeks to years (as commonly performed in perinatal cell banks for clinical use). Upon thawing, the cryopreserved cells were analyzed for surface markers and transcriptome analysis. The present data support the consistency of the cellular hAEC product, before and after cryopreservation. While there was approximately a 12% loss of viability upon thawing, the cells recovered after cryopreservation maintain the same surface markers as the cells prior to cryopreservation. Constitutive presence of epithelial markers, such as CD49f and CD326, showed that the isolation procedure is highly selective for epithelial cells, and that they nicely tolerate the cryogenic procedure. Thawing and subsequent wash steps resulted in an average loss of 35% of the cells initially frozen. 

As expected, the endothelial marker, CD31, was absent from all preparations examined. Since the amnion is an avascular membrane, vascular endothelial cells would not be expected to be present in isolates of hAEC cells. The presence of passenger (maternal) hematopoietic cells in a hAEC preparation could elicit an allogeneic immune reaction if transplanted, and sporadic preparations of hAEC showed up to 4% CD45 positive cells before cryopreservation. However, only five of 14 preparations contained remaining CD45 cells upon thawing, and in an extremely low amount (1% or less). 

Other surface markers present in some preparations were the mesenchymal markers: CD105 and CD44, which represented up to 5% (CD105) or 3% (CD44) in different preparations, before the cryogenic procedure. After cryopreservation, CD44 positive cells were absent in 12/14 cases and CD105 positive cells were only detectable in four of 14 cases, and sporadically (one case only) had more than 1% CD105 positive cells. It is not even clear that residual MSC in an hAEC preparation would be detrimental since the immunomodulatory properties of MSC are similar to hAEC. A recent report of the transplantation of hAEC into pre-term babies with established bronchopulmonary dysplasia reported release criteria as cells with >96% epithelial markers and less than 1% of cells positive for CD45 or mesenchymal markers [19]. The present studies indicate that those product release criteria would have been met in 13 of 14 of the cases reported here.

Once it was determined that the hAEC recovery after cryopreservation was acceptable, attention was focused on the examination of the properties that are thought to be critical for hAEC engraftment and long-term function after transplant. 

In clinical studies, allogeneic hAEC were transplanted with no signs of rejection [18,19]. In immune-competent models and patients, hAEC cells engrafted and survived, without administration of immunosuppressive drugs [13,16,33]. The immunomodulatory properties of hAEC are a key feature that makes them attractive candidates for cell transplantation. The hAEC are often referred as ‘immune-privileged’ cells, since they lack the expression of HLA class II antigens (as depicted by HLA-DR staining in Figure 3) as well as co-stimulatory molecules [19,20]. Polymorphic histocompatibility complexes have been shown to be expressed at different levels on hAEC: a low-density expression of HLA-Ia molecules is commonly detected on hAEC [20]. Recently, a constitutive expression of non-canonical, oligomorphic HLA class Ib molecules on hAEC was reported [21]. 

HLA-G molecules play a vital role in protecting the fetus, a ‘semi-allograft’ tissue from the mother’s immune system and can induce apoptosis in activated T cells and also modulates the activity of the natural killer cells (NK) by binding to specific receptors CD85d, CD158d, and CD85j, respectively [34,35]. Like HLA-G, HLA-E and HLA-F have potent immunomodulatory properties, although less is known about their mode of action. Transcriptome analysis showed expression of all non-polymorphic molecules (HLA-E, -F, and -G) by hAEC. 

In addition to the HLA class Ib molecules, other immunomodulatory functional enzymes, previously described on MSC, have been recently identified on hAEC [25]. Plasma membrane nucleotidases are cell surface enzymes, which represent functional bridges between cell cytoplasm and the environment. Purinergic mediators, such ATP and adenosine released into the extracellular space, have a role in proliferation and angiogenesis, and contribute to immune cell function by the inhibition of dendritic, T, and NK cells, and the promotion of T regulatory cells and M2 macrophages. Once in the extracellular environment, ATP is hydrolyzed into adenosine, by ecto-nucleotidase CD39 and CD73 in a tightly regulated process [36]. Alternatively, adenosine is produced by metabolizing NAD^+^, and this alternative pathway is initiated by CD38 and CD203a and culminates in CD73 [37]. All of these ectoenzymes play a role in modulating immune effector cells [25]. Previous reports have suggested that the ecto-nucleotidases operate in a discontinuous fashion, where different receptors are expressed on different cell types, resulting in a synergic effect based on cell-to-cell interactions. Human amnion epithelial cells were the first cell population reported to express all of these functional ectoenzymes on a single cell type, to drive the production of adenosine via canonical and alternative pathways [25]. Additionally, all these surface enzymes were maintained after cryogenic procedure.

Stem cell marker genes were investigated, since expression of these pathways will likely contribute to proliferation and self-renewal of hAEC as well as determine their ability to differentiate to different cell types [4]. The expression of SOX2, OCT4, and NANOG was quite consistent in the 14 cases examined. High levels of DLK1 were observed in all cases. It was reported that when transplanted into the liver, DLK positive precursor cells differentiate into hepatocytes [38] and hepatic differentiation of hAEC has been reported by us and others [5,13,14,15,16,23]. While embryonic stem cells express many properties of multipotent or pluripotent stem cells, they are different in important aspects from hAEC. Telomerase activity contributes to the regulation of proliferation, however telomerase reverse transcriptase (TERT) was not expressed by hAEC, suggesting a cell type with more limited proliferation capacity than other stem cells [39]. This may be an important safety issue by reducing the possibility of unlimited growth and possible tumor formation. 

The matrix metalloprotease (MMP) and tissue inhibitors of matrix metalloprotease (TIMP) play important roles in cell migration, invasion, tissue remodeling, and proliferation [40,41]. Cell migration and remodeling of matrix is also a necessary step in engraftment of cells following transplantation [42]. Breakdown of extracellular matrix results in the release of growth factors that were bound to ECM and makes them available to cellular receptors [40]. The hAEC predominantly express MMP-2, -3, and -9 in all cases analyzed. Both MMP-2 and -9 degrade collagens and play a pivotal role in mobilization, homing, and engraftment of hematopoietic stem/progenitor cells as well as other cell types [41]. Substrates for MMP-3 include collagens, fibronectin, laminin, and elastin and MMP-3 can also activate MMP-9. Active MMPs are inhibited by the tissue inhibitors of metalloproteinases (TIMPs) [41,43]. The balance between MMPs and TIMPS such as TIMP-1 are important in regulating MMP activities [40]. Further studies will evaluate and determine the functional activity and ECM remodeling activity performed by different batches of primary hAEC.

There are now two reports describing the transplantation of amnion-derived cells in patients affected by life-threating disorders. Five patients with Niemann–Pick disease type B were treated with amnion-derived cells [44], and recently six premature babies with bronchopulmonary dysplasia received hAEC treatment [19], confirming the strong immunomodulatory and regenerative capacity of hAEC. Our group has extensive experience with liver cell-based therapy of acute, chronic, and congenital liver disorders in patients [30,31]. Encouraging evidence of hepatic maturation by implanted hAEC in immune-proficient animals led our group to obtain approval to transplant allogenic hAEC into patients without the administration of immunosuppressive drugs. The recipient candidate are patients otherwise qualified for hepatocyte transplant. Data contained in this report are an important step in the collection of preclinical data necessary to proceed with manufacturing and banking of hAEC for clinical use. Surface marker expression analysis indicates that the population of hAEC recovered after cryopreservation is highly enriched for hAEC with minimal inclusion of other cell types (less than 1%). The expression of many pathways necessary for engraftment and long-term survival of hAEC following transplant procedures, including stem cell, immunomodulatory, and matrix remodeling pathways are maintained in cryopreserved hAEC. 

## Figures and Tables

**Figure 1 cells-09-01696-f001:**
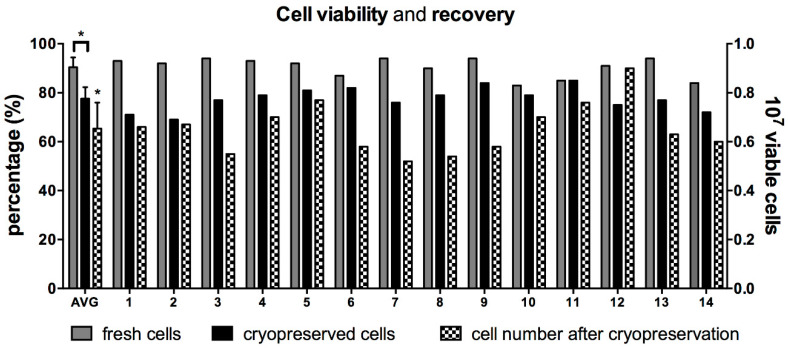
Cell viability and recovery after cryopreservation. (A) Cell viability of human amnion epithelial cells (hAEC) isolated from 14 different full-term human placentae are represented as grey bars. Similarly, hAEC viability after cryopreservation is represented as black bars. The first bars on the left represents average viability ± standard deviation (AVG). Cell recovery for every case and average are shown as black and white, checkered bars. * *p* < 0.001, *t*-test on paired values.

**Figure 2 cells-09-01696-f002:**
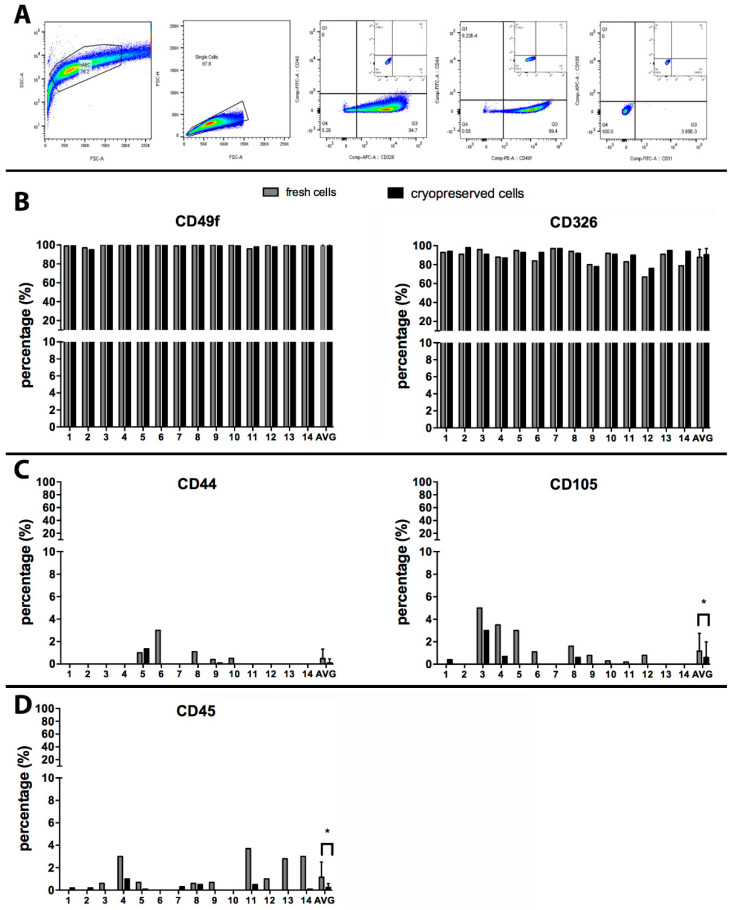
FACS analysis on fresh and cryopreserved hAEC: (**A**) a representative gating strategy is represented in panel A, where forward and side scatters identified intact cells, an additional gate selects single cells, later gated for different surface markers using two different fluorochromes. The negative isotype control for the respective fluorochromes is shown in the small dot plot graph. (**B**) Cells positive for CD49f or CD326 before (grey bars) and after cryopreservation (black bars). (**C**) Amnion-derived mesenchymal stromal cells (MSC) were identified based on their expression of characteristic markers (CD44 and CD105), before (grey bars) and after cryopreservation (black bars). (**D**) The hematopoietic marker CD45 was measured on cells in suspensions obtained after isolation (grey bars) and after cryopreservation (black bars). All values measured in 14 preparations are reported, with an average + SD as the last bar on the right. * *p* < 0.001, *t*-test on paired values.

**Figure 3 cells-09-01696-f003:**
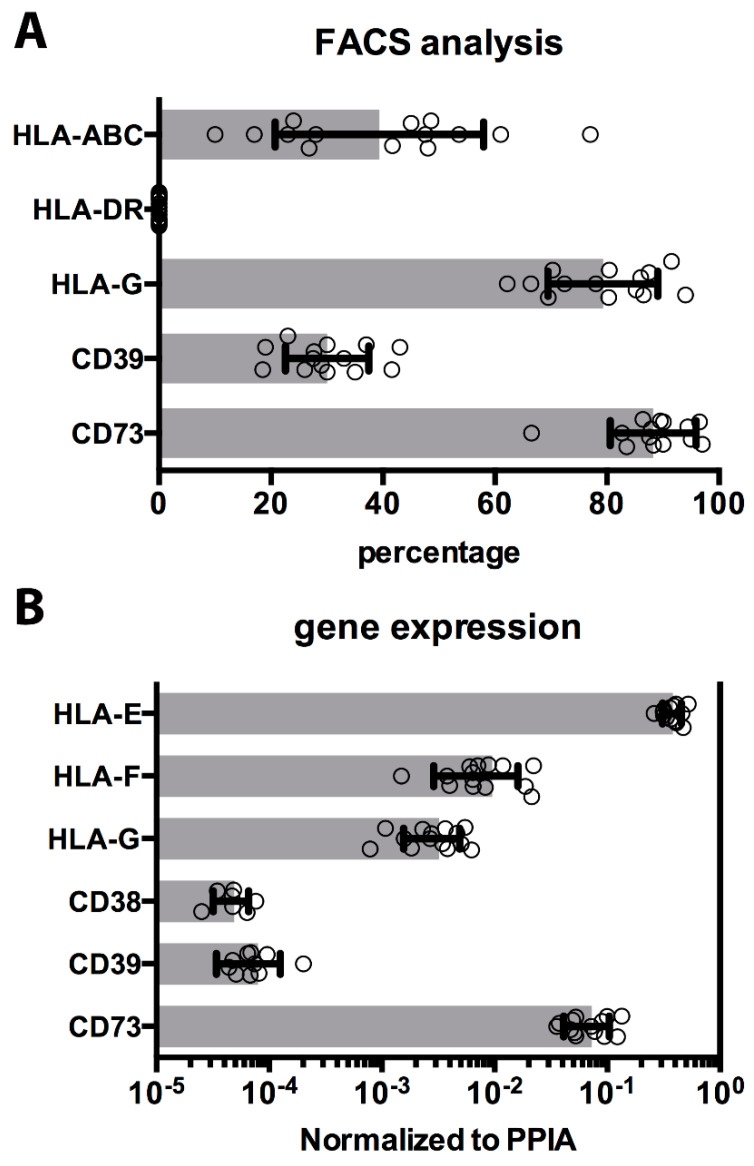
Immunomodulatory molecules on cryopreserved cells. (**A**) The percentage of cells positively detected by FACS for surface molecules as HLA classes Ia, Ib, and II, and ecto-enzymes CD39 and CD73 are shown as horizontal histograms with average values ± SD, and values (dots), measured in all 14 cryopreserved cases. (**B**) Gene expression analysis similarly evaluated and confirmed HLA-Ib molecules and ecto-nucleotidases expression. The results are shown as histograms with average values ± SD, and values (dots), measured in all cryopreserved hAEC cases.

**Figure 4 cells-09-01696-f004:**
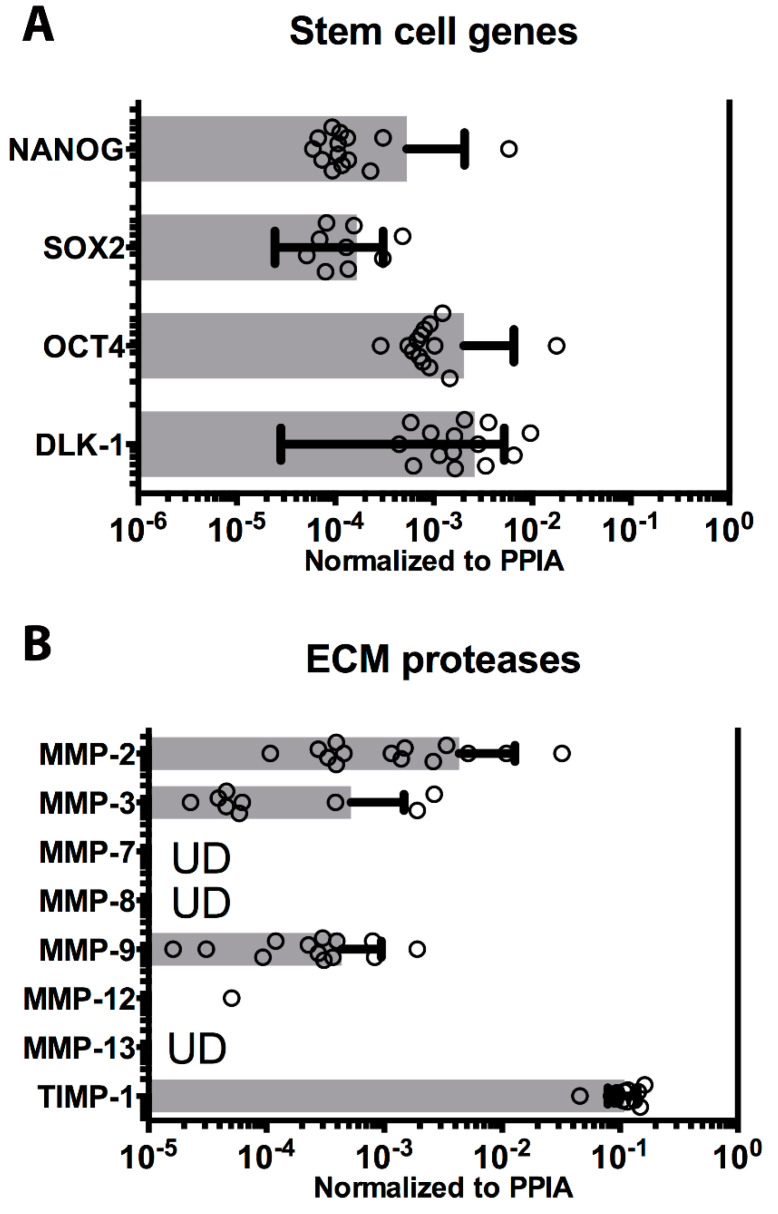
Stem cell transcription factors and extracellular matrix (ECM) protease expression on cryopreserved cells. (**A**) The level of expression of stem cell markers, commonly measured in embryonic stem cells, is shown as horizontal histograms with average values ± SD, and values (dots). All four genes SOX2, OCT4, NANOG, and DLK-1 were expressed in all 14 batches of cryopreserved hAEC cases. (**B**) The matrix metalloprotease (MMP-2, -3, -7, -8, -9, -12, and -13) and tissue inhibitor of metalloproteinases (TIMP)-1 level of expression were measured in all hAEC cases and shown as horizontal histograms with average values ± SD, and values (dots) when above limits of detection. UD indicates gene measured at undetectable level.

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
