# Peer review of "Effects of Cryogenic Storage on Human Amnion Epithelial Cells"

_cells, 2020, doi:10.3390/cells9071696_

Round 1

Reviewer 1 Report

The authors sought to understand how human amnion epithelial cells (hAEC) are affected by cryopreservation. As these cells are being investigated for their potential clinical use, the questions posed by the authors are logical. However, in order to truly add impactful and pivotal knowledge to the field, this reviewers feels that additional work must be undertaken. 

  1. The authors only one formulation cryopreservation agent. Do all cryopreservatives affect hAECs in the same manner? This would be useful for others in the field to know, particularly in relation to commercially available formulations. Looking at a single formulation for cryopreservation provides limited information to the readership.
  2. The authors comment on the impact of cryopreservation on hAEC markers and acknowledge the functional roles of these cell surface proteins e.g. ecto-enzymes, but did not actually test the impact of cryopreservation on functional differences.
  3. Given that MMPs exist as proenzymes, the MMPs and TIMPs would have been much more convincing on zymograms or enzymatic assays rather than gene expression data, which are not indicative of the status of the cells following cryopreservation

Author Response

The authors sought to understand how human amnion epithelial cells (hAEC) are affected by cryopreservation. As these cells are being investigated for their potential clinical use, the questions posed by the authors are logical. However, in order to truly add impactful and pivotal knowledge to the field, this reviewers feels that additional work must be undertaken.

We thank the Reviewer for finding time to read and suggest new approach that may help us in improving current and future study of ours.

  1. The authors only one formulation cryopreservation agent. Do all cryopreservatives affect hAECs in the same manner? This would be useful for others in the field to know, particularly in relation to commercially available formulations. Looking at a single formulation for cryopreservation provides limited information to the readership.

We thank the Reviewer for this comment. The primary scope of current study is to validate cryopreserved product in comparison with freshly isolated cells. So far, cryogenically preserved cells have been largely used in allogenic cell-based therapies, starting from cord blood hematopoietic stem cells to the recent infusion of hAEC (ref #19: Lim et al, Stem Cells Transl Med 2018). As the Reviewer correctly pointed out, several commercial cryogenic solutions are currently offered, and few selected are GMP-grade. All the formulations have in common the use of DMSO as cryoprotectant. Thus, our group has been able to test all the commercially available solutions on primary hAEC. Nevertheless, we thought such comparison may be considered beyond the scope of the current study, where multiple batches of primary hAEC have been isolated and cryopreserved using a protocol and reagents revised and optimized for cGMP manufacturing process (ref #27: Gramignoli et al. Curr Protoc Stem Cell Biol 2016). However, as we mentioned before, such comparative analysis has been performed and here below are the unpublished results. We include a graph with original data supporting the use of different commercial GMP-grade solutions to cryopreserved hAEC. We can also consider to include such analysis in the manuscript is strictly required, aware that may complicate the take-home message for the reader and expanding discussion and scope for the current study. A second manuscript is in preparation to properly address issue behind the use of different cryogenic solutions in clinical programs worldwide, and we would respectfully save for such comparative study.

  1. The authors comment on the impact of cryopreservation on hAEC markers and acknowledge the functional roles of these cell surface proteins e.g. ecto-enzymes, but did not actually test the impact of cryopreservation on functional differences

We would like to thank reviewer for the opportunity to clarify this issue. We would like to take the opportunity to state that ectoenzymes are ubiquitarian molecules, expressed in almost all the cells of human body. However, so far, only primary hAEC have been proved to express a homogenic complete set of adenosinergic ectoenzymes (as reported in ref #25: Morandi et al. J Immunol 2019). The production of ADO and relative immune-modulatory activity played by primary cryopreserved hAEC has been largely described in our previous study (ref. #25). Additional analysis on functional role of additional plasma membrane enzymes are the object of ongoing studies. 

  1. Given that MMPs exist as proenzymes, the MMPs and TIMPs would have been much more convincing on zymograms or enzymatic assays rather than gene expression data, which are not indicative of the status of the cells following cryopreservation

We are extremely grateful to the reviewer for such important suggestion. As stated before, additional functional studies are ongoing to properly evaluate and investigate functional role for different plasma membrane molecules as well as secreted mediators. As suggested by the reviewer, an enzymatic assay (Human Cell Protease/Protease Inhibitor Profiler Array Kit) has been order to further analyze secreted matrix metalloproteases (MMP) and tissue-inhibitor of metalloproteases (TIMP). The constitutive expression of MMP molecules has been largely described in previous publication (some cited in the current manuscript, refs #40-42), thus we considered important to evaluate such expression in our cryopreserved batches. As we stated at page 11, we recognized the importance that secreted peptides like MMPs and TIMPs have in cell migration and engraftment. However, the functional role and efficiency in remodeling ECM performed by primary hAEC is beyond the scope of current study. We are conducting more specific evaluations on a different set of cells, to profile and simultaneously detect the relative changes of a large plethora of protease inhibiting analytes in hAEC isolated from human placentae. As consequence, we respectfully consider to include such large set of data and related analysis in extracellular matrix remodeling into a second manuscript, where the scope is not to compare fresh vs cryo hAEC, but we intend to detail and profile secreted proteases, critical in some regenerative medicine application for selective human tissues.

However, we thought it was important to mention the limitation in current study, As Reviewer suggested, and need of further analysis for such critical soluble enzymes, thus we included (and highlighted in red for revision) the sentence “Further studies will evaluate and determine the functional activity and ECM remodeling activity performed by different batches of primary hAEC” in the Discussion section (page 12).

Reviewer 2 Report

Availability of stem cells of defined and unchanged phenotype is crucial for the success of cell-based therapies in regenerative medicine. In this aspect, cell banking can provide stable batches of therapeutic products in respect of current good manufacturing practice (cGMP). Based on the growing amount of studies describing perinatal stem cell advantages in translational research, the current manuscript evaluates the effect of cryopreservation on hAEC in comparison with freshly isolated cells from the same placentae. Considering the large field of clinical applications provided by the hAEC, the evaluation of cryopreservation on cell viability and function appears as a mandatory pre-requisite before moving to systematic use of banked cryopreserved cells.

The authors have well recognized expertise in the field.

The study by Raghuraman Cet alt. evaluates the recovery rate, viability, and changes in expression of several markers of the batch purity and phenotype of hAECs before and after cryopreservation. The main strength of this study is the exploration of changes in marker expression, which could impact further cell engraftment and function in vivo. The investigations are well conducted with defined methodology and clear results. Nevertheless, some modifications and additional explanations are needed to improve the manuscript quality.

Major comments:

  • It is interesting to know, what were the amnion acceptance criteria for cell isolation? This information should be included in the material and method section.
  • Authors should indicate cell yield per isolation (how many cells were obtained per membrane) for each preparation.
  • In the materials and methods, authors state that cells were thawed in FBS-supplemented buffer. Authors should discuss how the use of FBS complies with GMP regulations.  
  • The FACS protocol used in the study involves fixation of the cells before analysis, which can have a negative effect on cell surface marker detection (doi : 1371/journal.pone.0068519, doi : 10.1002/cyto.a.20392). Did the authors check if the fixation does not impact negatively the cell surface marker expression? Did they compare it to a protocol performed on fresh, unfixed cells? Why did the authors choose to fix the cells, when performing FACS analysis on unfixed cells would be more in line with the clinical configuration?
  • Figures 3 and 4 show expression of immunomodulatory molecules, stem cell transcription factors and ECM proteases by hAECs only after cryopreservation, data for freshly isolated cells are not shown. It would be interesting to see these data on fresh cells in order to demonstrate the impact of the cryopreservation on these markers. This is especially important for the gene expression of stem cell transcription factors and matrix metalloproteases. It has been shown that MMP expression is linked to Epithelial to Mesenchymal transition (DOI: 10.1007/0-387-28671-3_20), which is related to a decrease in hAEC immunomodulatory properties (doi: 10.1038/s41598-017-03908-1.). Moreover, MMP are overexpressed in most tumors, and thus a change in MMP expression induced by cryopreservation could involve critical safety issues in the clinical use of cryopreserved hAEC.
  • Was TERT expression undetectable both before and after cryopreservation?

Minor details:

  • Abstract – line 15 : a space is lacking between « from » and « 14 different donors »
  • Introduction – line 46 : [9] refers to Type 1 Diabetes (not type 2)
  • Introduction – line 64 : safety (not safely)
  • Figure 1 and 2 : Dot plots indicating individual data for each cell preparation as well as mean and SD would be easier to understand (as in figures 3 and 4) than the current form of the bar charts.

Author Response

Availability of stem cells of defined and unchanged phenotype is crucial for the success of cell-based therapies in regenerative medicine. In this aspect, cell banking can provide stable batches of therapeutic products in respect of current good manufacturing practice (cGMP). Based on the growing amount of studies describing perinatal stem cell advantages in translational research, the current manuscript evaluates the effect of cryopreservation on hAEC in comparison with freshly isolated cells from the same placentae. Considering the large field of clinical applications provided by the hAEC, the evaluation of cryopreservation on cell viability and function appears as a mandatory pre-requisite before moving to systematic use of banked cryopreserved cells.

The authors have well recognized expertise in the field.

The study by Raghuraman Cet alt. evaluates the recovery rate, viability, and changes in expression of several markers of the batch purity and phenotype of hAECs before and after cryopreservation. The main strength of this study is the exploration of changes in marker expression, which could impact further cell engraftment and function in vivo. The investigations are well conducted with defined methodology and clear results. Nevertheless, some modifications and additional explanations are needed to improve the manuscript quality.

We thank the Reviewer for finding time to read and pinpoint all the inaccuracies that need to be clarified in order to improve the quality of our study.

Major comments:

  • It is interesting to know, what were the amnion acceptance criteria for cell isolation? This information should be included in the material and method section.

We thank the Reviewer for such comment. All the placenta tissues utilized in the current study have been collected according procedure in place at our cord blood unit. Specifically, we routinely test mother-to-be for several pathogens and perform hematological analysis the day before c-section procedure. Accordingly to Reviewer’s request, we included (and highlighted in red for revision) the following details in Materials and Methods (page 2): All the pathogen-positive deliveries (including HBV, HCV, syphilis and HIV) were excluded, and current revised exclusion criteria includes positiveness for COVID-19. No information regarding human donors (mother and newborn child) has been transmitted or diffused, and the details have been de-personalized, in order to respect privacy. Placentae were delivered and processed within 3 hours post-partum.

  • Authors should indicate cell yield per isolation (how many cells were obtained per membrane) for each preparation.

We appreciate Reviewer’s comment regarding cell yield.  We agree that hAEC recovery from primary human amnion membrane is an important and critic parameter for future clinical use. As we detailed in our hAEC isolation protocol (ref #27: Gramignoli et al. Curr Protoc Stem Cell Biol 2016), our optimized procedure allows us to extract up to 300 million epithelial cells from a single amnion membrane. We are aware that a considerable variability in cell yield between preparations has been described, generally ascribed to the quality of donor tissue. As we stated in our previous publication, we consider a successful isolation when it generates a yield of more than 5 million hAEC per gram of processed tissue. Hence, as requested by the Reviewer, we stated cell yield as first parameter in our Results section (page 4) and highlighted in red for further revision: Fourteen human placentae have been collected and generated cell suspensions characterized by limited variability (16 ± 7 million viable cells/gr of processed tissue).  

  • In the materials and methods, authors state that cells were thawed in FBS-supplemented buffer. Authors should discuss how the use of FBS complies with GMP regulations.

We would like the Reviewer for catching this critical detail: despite the fact animal sera have been largely used as supplements for cell manipulation before infusion in previous clinical application, current requirement encourages removal of such additive, preferably substituted with human products. Our thawing procedure has been revised and FBS substituted with human albumin, and such step validated using same cell batches here described and more. To further elucidate such critical step and dispel any doubt regarding the accomplishment with current GMP requirements, we added the following sentence at page 10: The formulation of reagent to thaw hAEC has originally included additive of animal origin (FBS). Such supplement has been replaced with human albumin in revised procedure, and matched reliability and efficiency (as validated in several hAEC batches, including cases here analyzed, data not shown). In later study, we have described corroboration in using human albumin as additive for thawing solution, and we would be happy to include such comparison in supplementary table if required.

  • The FACS protocol used in the study involves fixation of the cells before analysis, which can have a negative effect on cell surface marker detection (doi : 1371/journal.pone.0068519, doi : 10.1002/cyto.a.20392). Did the authors check if the fixation does not impact negatively the cell surface marker expression? Did they compare it to a protocol performed on fresh, unfixed cells? Why did the authors choose to fix the cells, when performing FACS analysis on unfixed cells would be more in line with the clinical configuration?

We would like to thank reviewer for such valuable comment. We are aware that the use of fixatives has been described causing significant decrease in morphometric parameters (forward and side scatter) when blood leukocytes immunophenotyped (Stewart JC et al. cytometry 2007). As a consequence, gate adjustment is required and autofluorescence properly menaged and considered. We believe that the use of isotype controls in all samples here analyzed (shown in Fig.2), and the equity in sample manipulation (Ab staining procedure and fixative exposure) can efficiently exclude any bias in comparison between fresh and cryopreserved samples. Unfortunately, the use of fixative is mandatory in our Institute, for safety reason. We previously performed comparative analysis on alive cells and PFA-treated samples, reporting similarities and lack of difference in a population of (epithelial) cells characterized by high dimension and characteristic morphometric parameters (Gramignoli R, et al. Methods in Bioengineering: Cell Transplantation (2011). Edited by: Alejandro Soto-Gutierrez, Nalu Navarro-Alvarez, Ira J. Fox. ISBN 978-1-60807-015-2; Chapter 12; pp183-98). We are confident the Reviewer will agree in considering fixative as uninfluential additive in our analysis, where equally treated samples (fresh and cryo) have been analyzed.

  • Figures 3 and 4 show expression of immunomodulatory molecules, stem cell transcription factors and ECM proteases by hAECs only after cryopreservation, data for freshly isolated cells are not shown. It would be interesting to see these data on fresh cells in order to demonstrate the impact of the cryopreservation on these markers. This is especially important for the gene expression of stem cell transcription factors and matrix metalloproteases. It has been shown that MMP expression is linked to Epithelial to Mesenchymal transition (DOI: 10.1007/0-387-28671-3_20), which is related to a decrease in hAEC immunomodulatory properties (doi: 10.1038/s41598-017-03908-1.). Moreover, MMP are overexpressed in most tumors, and thus a change in MMP expression induced by cryopreservation could involve critical safety issues in the clinical use of cryopreserved hAEC.

We would like to thank the Reviewer for allowing us to further discuss such important feature characterizing hAEC. As previously stated in clarification for Reviewer #1 concern, the amnion membrane has been described playing primary role in perinatal tissue remodeling during the final phases of the pregnancy, including leading role in membrane rupture and labor. Such critical events have been previously described as the result of structural alterations in ECM driven by MMPs and other proteases. Primary hAEC have been largely reported as constitutive secretors for MMP and TIMP, and the importance of such proteases for ECM remodeling and enhanced engraftment of donor cells has been largely proposed and recognized as added value to hAEC-based strategies.

Similarly, we have previously described characteristic presence of markers, commonly described on pluripotent stem cells, in freshly isolated hAEC (Miki T et al. Stem Cells 2005). Such expression is not affected by cryogenic preservation as detailed in the current study.

  • Was TERT expression undetectable both before and after cryopreservation?

We are glad to confirm Reviewer absence of TERT expression in freshly isolated cells, as our group’s seminal manuscript detailed in 2005 (Miki T et al. Stem Cells 2005). Here we report confirmation, reinforcing the positiveness in safety, confirmed by two first-in-human studies performed so far (and cited in the manuscript as refs #19 and #44)

Minor details:

  • Abstract – line 15 : a space is lacking between « from » and « 14 different donors »

Thanks for noticing that, the mistypo is now fixed.

  • Introduction – line 46 : [9] refers to Type 1 Diabetes (not type 2)

Thanks for highlighting the mistake. We have corrected type 2 in type 1 diabetes

  • Introduction – line 64 : safety (not safely)

Thanks again, mistypo corrected

  • Figure 1 and 2 : Dot plots indicating individual data for each cell preparation as well as mean and SD would be easier to understand (as in figures 3 and 4) than the current form of the bar charts.

We partially agree with this comment. We edited the graphs in Figure 1 and 2, but unfortunately, due to limited size, the graph will be quite difficult to read and the dots may be quite confusing. Respectfully, we would like to keep the graph in the original form and the SD bar elucidates lack in significant variability.

Round 2

Reviewer 1 Report

The authors have elected not to address the concerns raised previously and this is rather disappointing as this severely limits the utility of the manuscript and contribution to the current knowledge base on the hAEC. It appears that all requests for additional data are considered beyond the scope of this study, making this study of limited use to the cell therapy community. The manuscript then fails to address the primary aim of investigating the effect of cryopreservation on hAEC quality as outlined in the final paragraph of the introduction.